# Restoring Activity of Milk Thistle (*Silybum marianum* L.) on Serum Biochemical Parameters, Oxidative Status, Immunity, and Performance in Poultry and Other Animal Species, Poisoned by Mycotoxins: A Review

**DOI:** 10.3390/ani13030330

**Published:** 2023-01-17

**Authors:** Alessandro Guerrini, Doriana Eurosia Angela Tedesco

**Affiliations:** Department of Environmental Science and Policy, University of Milan, Via Celoria 10, 20133 Milan, Italy

**Keywords:** silymarin, phytoextract, bioactive compounds, feed additives, animal health

## Abstract

**Simple Summary:**

Mycotoxins are secondary metabolites produced naturally by toxigenic fungi, which elicit a toxic response in humans and animals. The livestock sector is showing a greater focus on phytoextracts for their limiting and controlling the toxic effects of mycotoxins with safeguarding animals’ health and performance. Milk thistle and its bioactive compound silymarin are considered useful for this scope. This review underlines the efficacy of milk thistle to counteract mycotoxins toxicity on organs, biochemical and immunological functions, and performance in poisoned poultry and livestock. The use of milk thistle as a whole plant, seed, and its standardized silymarin extract in mycotoxicosis cases produce positive effects on the maintenance of the animals’ performance, restoring liver functionality due to its known hepatoprotective and antioxidant effects, reducing organ lesions caused by intoxication. The use of milk thistle in animal farming can be useful since the bioactive compounds, also if present in variable amounts, can help the animals to counteract the effects of mycotoxins. The use of silymarin, due to its cost, can be useful if it reported the specific bioactive compounds it contained.

**Abstract:**

Grains are major farm animals’ diet ingredients, and one of the main concerns is when are mycotoxin (MyT) contaminated, compromising animals’ health, performance, and product safety. Among the natural phytocompounds that are being used to prevent MyT damage, silymarin (SIL), an extract from the seed of the milk thistle (MT) is a suitable candidate. This review aims to examine the scientific evidence concerning the anti-MyT toxicity effects of MT/SIL in poultry and livestock. In vitro and in vivo studies (*n* = 27) showed that MT whole plant, seed, or SIL-standardized extract had positive effects on animal health, performance, and restoring the hepatic activity, with a reduction of organ lesions caused by MyT. Furthermore, showed utility for combating MyT-immunodepression, improving intestinal health, and limiting the excretion of toxins residues in food of animal origin, although in some cases, MT/SIL supplementation does not produce appreciable effects. The use of MT in animal nutrition can be useful since the bioactive compounds, also if present in variable amounts, can help the animals to counteract the effects of MyT. The use of the phytoextract of SIL, due to its cost, can be useful if it reported the specific bioactive compounds, recognize for their pharmacological activities.

## 1. Introduction

Mycotoxins (MyT) are secondary toxic metabolites naturally produced by fungal species, that contaminate agricultural commodities before or under post-harvest conditions, and during their storage [1]. There are about 200 species of MyT-producing fungi [2], but *Aspergillus*, *Fusarium*, *Penicillium*, *Alternaria*, and *Claviceps* genres, are the main ones responsible for the production of MyT [3,4]. The most important MyT for the safety of animal feed and human food, include aflatoxins (AFs), and relatives isoforms (i.e., AF-B_1_, B_2_, G_1_, G_2_, and M_1_) produced primarily by *A. flavus* and *A. parasiticus* [5], fumonisins (FUM-B1, B2), ochratoxins (OT-A, B, C) produced by *A. niger*, *A. verrucosum*, and *A. ochraceus*, zearalenone (ZEA/ZON), Ergot sclerotia, patulin, citrinin, and deoxynivalenol-DON, T-2, two trichothecenes toxins [6,7]. The *Fusarium* genus and, to a lesser extent, other genera such as *Beauveria* and *Isaria*, are responsible for the fumonisins production, but also beauvericin and enniatins, two emerging MyT [8]. Some of these MyT are considered the most toxic for animals, as well as for humans, where dietary exposure can have severe effects, including carcinogenicity [9]; this is the case of AFB_1_, the most toxic of its group. The International Agency for Research in Cancer (IARC) and World Health Organization (WHO) classified it as a highly toxic compound and highly carcinogenic to humans (Group 1) [10] due to pro-carcinogen metabolite product by the activation of hepatic microsomal cytochrome P450 (CYP450), the AFB_1_-8,9-epoxide [11]. Otherwise, OTA and FUM are classified as possibly carcinogenic to humans but ascertained carcinogenic to animals (Group 2B) [10,12], while DON and T-2, derived from *F. sporotrichioides*, are not classifiable as carcinogens to humans (Group 3) [10]. 

Cereals such as corn, wheat, and also soy (leguminous as a source of protein), which are the main components of daily livestock and poultry diets, can be easily contaminated with fungi-producing MyT. Animal susceptibility to acute and chronic toxic effects, in particular for aflatoxicosis, can vary widely [13], and also depends upon the structure of the toxin involved, its route, and time of exposure [14]. MyT are metabolized in the gastrointestinal tract, liver, or kidneys, under their chemical properties. AFB_1_, OTA, FUM, DON, and T-2 toxins significantly affect poultry health and productivity [15]. Poultry flocks suffering from mycotoxicosis showed a worsening in their health, and performance (body weight-BW, feed conversion ratio-FCR, carcass weight, eggs weight, etc.) [16], associated with immunosuppressive effects, widespread damage to organs (in particular hepatotoxicity and nephrotoxicity), neurotoxicity, carcinogenicity, and high mortality [4,17,18,19,20]. The control of MyT production and decontamination is crucial for farmers to minimize economic loss. Nonetheless, it should not be overlooked that the impact of climate change has been identified as an emerging issue for food and feed safety [21], and that has an impact, for example, on the presence of AFs in some crops. In Europe, increasing temperatures based on an increase of +2 °C in the climate change scenario would increase the probability of AFs contamination from low to medium in European countries [21]. For several years, the studies regarding the use of substances for reduction of the negative effects of the contaminated feed by MyT, have understood the use of toxins binder (TB), adsorbents (i.e., beta-glucans, zeolite, bentonite, hydrated sodium calcium aluminosilicate, etc.), with the scope to suppress or reduce the absorption and/or promote the MyT excretion [6,19,22]; medicinal herbs and their products (i.e., essential oils, whole plant, seeds, etc.), exploiting the pharmacological actions of their phytocompounds, for reducing the negative effects of MyT [23,24]. These strategies are focused to protect the target organs such as the liver and strengthen the immune system of animals to maintain their health and performance.

Regarding the latter strategy, milk thistle (MT), *Silybum marianum* L. (Gaertn.), a plant member of the Asteraceae family known as a potent hepatoprotector mainly thanks to its bioactive compound: silymarin (SIL). This compound consists of different flavonolignans, and the minimum content of SIL in mature MT fruit is 1.5% (expressed as silibinin) [25], and often ranges between 1–8% of dry matter [26]. SIL contains flavonolignans such as silybin (50–60%), isosilibinin (~5%), silicristin (~20%), silidianin (~10%), and other components such as taxifolin (~5%) [27]. Most of the beneficial effects of SIL have been attributed to its predominant component, silybin. Despite its low bioavailability, scientific evidence continues to highlight its biological relevance. However, due to the complexity of the absorption, metabolism, and disposition of various SIL-flavonolignans, it is still unclear which form (i.e., the parent flavonolignan or its metabolite/s) contributes to the overall effects in the body [28]. 

The pharmacological potential of MT is well documented by existing data that confirm the safety and tolerability of its herbal preparations in various settings related to hepatic disorders, and in farm animals to support their performance and health [29]. Detailed information on the multifunctional and multitarget activity and the potential mechanism of action of MT and derivative products were reported in the European Medicines Agency (EMA) report [25] and by Tedesco and Guerrini [29]. In the liver, SIL suppresses the increase in the hepatic enzymes, such as ALT, AST, and ALP, during liver damage, associated with the ribosomal RNA synthesis induced by SIL, improving liver regeneration and preventing the transformation of stellate hepatocytes into myofibroblasts [30,31,32], conducing to the restoration of normal liver functions and enzymes production at a normal value [33,34]. These main other properties, regarding the antioxidant and anti-inflammatory activity, are due to SIL’s capacity to act as both free radical scavenging (reactive oxygen species-ROS), and lipid peroxidation (i.e., malondialdehyde-MDA) inhibitors [35]. It acts, in fact, through the suppression of pro-inflammatory signals, derived from nuclear factor-κB (NF-κB) activation, involved in the induction of the synthesis of cytokines such as tumor necrosis factor-α (TNF-α), interleukins (IL-1, IL-6), and granulocyte-macrophage colony-stimulating factor (GM-CSF) [36,37]. To date, for this effect, SIL can be useful in the treatment of poisoned episodes in animals caused by drugs, heavy metals, pesticides, and toxins, including MyT, that negatively involve liver functionality. SIL interferes with the activity of CYP450 in the cells, reducing its activation system [38,39]. The same mechanism of detoxification can occur for AFs [40]. In the case of AFs poisoning, SIL limits the AFs damage interfering with CYP450 activity, so reducing epoxide production, as also demonstrated for other bioactive compounds, such as curcumin [41]. Additionally, the major AFs detoxification route is via conjugation of its AFs-epoxide to hepatic glutathione (GSH), catalyzed by the detoxification glutathione S-transferases (GSTs) enzyme. SIL would interfere positively with the regulation of this route, with the minor consumption of hepatic glutathione [42]. 

Various experimental studies were performed to demonstrate the hepatoprotective action of MT/SIL as a feed ingredient or feed additive, and their activity to restore health parameters and performance in farm animals, as previously described in “Use of Milk Thistle in Farm and Companion Animals: A Review” [29]. The present review aims to discuss the role and the effects of MT in MyT-poisoned animals in restoring the serum biochemical, and oxidative parameters, immunity, and performance, analyzing in vitro and in vivo studies. 

## 2. Research Methodology 

This review analyzes scientific papers reporting evidence of the use of MT and its phytocompound, the SIL, against MyT toxic effects. To explore the topic systematically in detail, a literature search was performed by entering keywords in specific scientific databases, including Minerva, PubMed (including ToxNet platform), Prime PubMed App, Medline Plus, Google Scholar, Scopus-Elsevier, Scifinder-n, Web of Science, ResearchGate and Scientific Information Database (SID). The Minerva database is the access point to the bibliographic resources available from the University of Milan.

The same papers consulted provided by the database search were also used for a cross-comparison, using the reference list cited. In this way, it was verified whether or not the databases also provided all the studies cited in the consulted papers. No specific timeframe of publication years or language was considered for the research in the databases. The search, using the same keywords, was performed two consecutive times at a distance of 1 month from each other, due to the continuous updating of the database. The literature search was performed from July to November 2022.

The search terms in each database consisted of the name of the animal species or zootechnical categories (e.g., chickens including *Gallus gallus*, broiler chicks, broiler chickens, broiler, rooster, pullet, hen, and laying hen), combined with the terms “milk thistle”, “*Silybum marianum*”, “silymarin”, and “silybin”, and the main MyT such as “AFB_1_”, “aflatoxin”, “OTA”, “ochratoxin”, “deoxynivalenol”, “DON”, “zearalenone”, “ZEA”, “Ergot sclerotia”, “patulin”, “citrinin”, and “T-2”, were inserted in each database.

The selection of the studies was conducted following specific criteria. To be included, publications had to provide at least one abstract written in English. The publications considered were in vivo studies including field trials on farms, experimental controlled trials (i.e., intoxicated-challenge animals), and in vitro/ex vivo studies on cells. Publications regarding case reports were not excluded. Instead, studies involving the use of toxin binders (TB), including the use of MT individually tested in combination with TB in the experimental trial, or with concomitant tests with other nutrients (i.e., vitamins, lycopene, etc.) were included. Additionally, considering the publications that investigated the effects of a mixture of different MyT. Contrarily, studies regarding the use of MT in a plant or extract mix were excluded.

## 3. Results from Databases

From the overall search on each database, with the use of the above-indicated keywords, the scientific publications found were 44. Excluding the same publications provided as duplicates from one or more databases, and applying the exclusion and inclusion criteria, the usefulness number of the final publications was 27. No publications relating to laying hens, turkeys, swine, goat, sheep, fish, and their all categories emerged from the research. Table 1, Table 2 and Table 3 summarize the data obtained and presented from the studies considered in chickens, other poultry species, and livestock, respectively. The following sections thus summarize the outcomes of identified experimental studies investigating the AFs toxicity and the effects of MT to contrast and restore the negative impact on production, biochemical and immunological effects in poultry and other livestock animal species.

### 3.1. Protective Effects of SIL Supplementation in Poultry against MyT Anatomopathological and Cyto-Histological Modification

As widely known and reported by several studies and scientific reviews, MyT intoxication in chickens produces adverse effects resulting in typical evident anatomopathological (macroscopic) and histological observations. However, the lesions and their severity vary based on chronic or acute exposure, but especially by MyT dose-dependent. An in vitro study reported that MT seed was able to diminish AFB_1_, in particular against its density and absorbency, probably due to its rich source of fiber able to bind the aflatoxin [43]. This could be directly useful to reduce the amount of MyT absorbed and metabolized and to maintain animal health. From the anatomopathological point of view, in an early study, broiler chicks fed with a contaminated diet with 0.8 mg/kg of AFB_1_ and SIL-phytosome, at a dose of 600 mg/kg-BW (SIL-Indena standardized extract) by gavage, it was found that liver weights were not different from the liver from the control group and the diet did not influence the organ’s weight nor restore it [40]. On the contrary, broilers fed with a diet supplemented with MT seed at a dose of 10 g/kg/diet after being exposed to AFB_1_ contaminated feed (80–520 μg/kg/diet), restored the negative effects of AFB_1_ on atrophy and the smaller weight of lymphoid organs such as the bursa of Fabricius, spleen, and thymus, counteracting the AFs immunosuppressive effect [44,45]. In quail, fed with diets containing 2.2 mg/kg of AFB_1_, at 35 days of age the testis proportional weight decreased, and the weight of the liver and spleen increased, damages restored with the supplementation of 1000 mg/kg/diet of SIL [63]. Conversely, Jahanian and colleagues [46], in AFB_1_-poisoned broiler denoted an increase in only heart weight, but not in the liver and other organs (including the pancreas and gallbladder), which were not restored by the supplementation with a dose of 500–1000 ppm/kg/diet of SIL (composition not reported) [46]. In broiler exposed to 3 mg/kg/diet of OTA, slight congestion and enlargement of the liver and kidneys with gallbladder distension was observed, which was associated with small hemorrhages on the epicardium and duodenal mucosa, and with catarrhal enteritis and hyperemia of the mucosal surface, without pathomorphological changes of internal organs [47]. Additionally, in this case, the lesions caused by OTA were not mitigated by the supplementation of a 1%/kg/diet of SIL for 32 days, at least macroscopically. The same observation was evidenced by Grizzle and co-workers [65], with the administration of 10–100 mg/kg-BW of SIL (SIL 80%, analytical composition not reported) in pigeons poisoned with 3 mg/kg-BW of AFB_1_, administered by gavage. The lesions caused by AFs such as macroscopic liver degeneration, necrosis, inflammation, and hemorrhage, were not alleviated by SIL. 

However, in most cases, even when the effects on organs were not macroscopically visible, histological evaluations are considered mandatory due to the anti-MyT effects of SIL on target organs, such as the intestine, lymphoid organs, liver, and kidneys. On the intestine, the morphometric analysis performed in a study conducted by Armanini and co-workers [48], it showed positive effects only in chickens fed with SIL alone, in a greater villus height and crypt depth, compared to the control and in MyT mix (AFB 0.05 ppm/kg/diet + FUM 20 ppm/kg/diet) plus SIL group (100 mg/kg/diet, commercial product 84.16% SIL-based). A lower intestine length was measured in broilers receiving AFB_1_ (500 ppb), and the MT administration improved the length of the ileum and duodenum plus jejunum when administered alone, at a dose of 1.0%/kg/diet, but not when administered combined with MyT [49]. The exposure to 2 ppm of AFB_1_, resulted in decreases in villus height (VH), villi height and villus height-to-crypt depth (CD) ratio (VH/CD), villi surface area, apparent villus absorptive area, muscular diameter in jejunal sections, increases in CD, goblet cell count (GCN) and lymphoid follicular count, at days 28 and 42 of age [46]. In this study, it was observed that the jejunal CD, GCN, lymphoid follicular count, and diameter were aggravated, as AFs levels increased. This effect was mitigated with the addition of SIL to the diet. Furthermore, at a maximum dose tested (1000 ppm/kg/diet) plus AFB_1_, a depression of the VH/CD ratio was observed [63]. In broiler poisoned by AFB_1_, a linear increase in CD, and a linear decrease in villus length (VL) and villus width (VW) was observed [50]. Despite this, also if the VL/CD ratio in the ileum was decreased significantly, the interaction between VW and GCN was unaffected by the consumption of AFB_1_ plus MT [46]. 

Histologically, a very in-depth investigation was conducted by Stoev and colleagues [47]. They evidenced that the pathomorphological alterations of the organs were seriously degenerative in chicks treated with OTA, at a dose of 3.0 mg/kg/diet, followed by those treated with OTA plus SIL (1%/kg/diet, 60% purity). In SIL-treated animals, the intestinal mucosa showed slight degenerative changes and depletion of cells in the lymph follicles, with a slight mononuclear cell infiltration. In the lymphoid organs, such as the bursa of Fabricius and the thymus, the main degenerative damages were found in chicks exposed to OTA, followed by those treated with OTA plus SIL (60% of purity). In the liver, the main target organs of the toxic effects of MyT, but also hepatoprotection of SIL, its addition revealed a reduction in cloudy swelling, granular or vacuolar degeneration of hepatocytes and a slight activation of the capillary endothelium and Kupffer’s cells in OTA-exposed chicks [47]. These data were also supported by in vitro study [51], where the cytotoxicity of OTA in primary chicken hepatocytes was investigated. Testing a dose of 1 µg/mL of OTA, the cell number and density decreased markedly and ruptured and necrotic hepatocytes confirmed that OTA induced a significant increase in apoptosis and related expression genes, with the activation of a mitochondrion-dependent pathway. When the cell cultures were treated with a dose of 0.1, 1, or 10 µg/mL of SIL (analytical composition not reported) for only 24 h, the cell viabilities increased, and there was a reduction in the apoptosis-associated genes expression, hepatic enzymes activity, and MDA level [51]. In a study conducted by Tedesco and co-workers [40], in the liver tissue of AFB_1_-treated chickens, were observed necrosis and multifocal portal infiltration of mononucleates, granulocytes, and eosinophils diffused in the parenchyma, especially at the portal areas. However, less severe lesions were evidenced when SIL-phytosome was administered. The same lesions in the parenchyma were observed when DON and ZEA were administered at a dose of 4.9 mg/kg/diet and 0.66 mg/kg/diet, respectively, in a high-sensitivity species, the ducks, for 47 days (chronic exposure) [60]. The lesions were represented by a high-grade vacuolar degeneration of hepatocytes cytoplasm, solitary hepatocyte necrosis, and also individual cell deaths of the mononuclear phagocyte system. Focal lymphocytic and histiocytic interstitial infiltrates and mild interstitial fibrosis was noted, nevertheless, these observations and lesions were milder in the treated group with MT seed (analytical composition not reported) at a dose of 0.5%/kg/diet [60]. At a lesser dose of AFB_1_ (3 mg/kg-BW) in pigeons’ liver tissue, necrosis with multifocal portal infiltration of mononucleates, granulocytes, and eosinophils, fatty infiltration, and vacuolization, bile-duct hyperplasia, cell necrosis, and inflammation were noted and were not alleviated with SIL (10–100 mg/kg-BW, SIL 80%, standardized extract) administration [65]. 

### 3.2. SIL Effects on Serum-Biochemical Parameters on MyT-Poisoned Poultry

The hepatoprotective activity of SIL induces different modifications in the production and activity of the hepatic enzymes and their correlated biochemical parameters. In a study conducted by Amiridumari and colleagues [52], MT seed supplementation (0.5–1%) to AFB_1_-contaminated feed (500 ppb/kg/diet) significantly increased serum glucose and decreased creatinine and AST levels, compared to AFB_1_ (500 ppb/kg/diet). At the same dose of AFB_1_, Fanni Makki and co-workers [53] denoted a significantly decreased AST enzyme but also uric acid, glucose, and gamma-glutamyl transferase (γ-GT) enzymes compared to the control group with the addition of 0.5–1% of MT seed. Similar results were obtained by Raei and co-workers [54] in chicks fed for 21 days with 2 mg/kg of AFB_1_, which showed a significantly declined Ca and HDL, as well as an increase in ALT and AST activity. Adding MT seed (10 g/kg/diet) enhanced the levels of Ca and HDL, reducing the activity of ALT and AST enzymes, without considerable variations in total serum protein, low-density lipoproteins (LDL), uric acid, and γ-GT levels. Even the addition for 41 days of a dose of 100 mg/kg/diet of SIL (SIL 84.16%) plus MyT mix including FUM + AFB, prevents the increase of ALT in broiler [48]. Evidence for this was also observed by Feshanghchi and colleagues [55], where MT seed powder (10 g/kg/diet: silichristin 2851 mg/kg, silybin B 8864 mg/kg, silymarin 100 mg/kg) significantly decreased AST and ALT activity. A significant decrease in AST activity was detected in growing quail fed with 500 mg/kg/diet of SIL, compared to the control, with no effects on ALT activity [64]. The effects of MT were investigated using the whole plant, compared to the effect of the seed administration, specifically on the influences of ALT enzyme activity. The addition of 0.5% MT seed powder, 1% MT plant powder, and 1000 mg/kg MT plant extract to the contaminated broiler diets with 500 ppb/kg/diet of AFB_1_, decreased ALT at 35 days of treatment, compared to the contaminated control [56]. At a dose of SIL even 20 times higher, the same effect was noted in quails, with the administration of 1000–2000 mg/kg/diet of SIL (analytical composition not indicated) plus AFs mix (2.2 mg/kg/diet, including AFB_1_ 68.19%, AFB_2_ 4.57%, AFG_1_ 24.96%, AFG_2_ 2.28%). SIL reduced ALT, AST, and ALP activity with a low value of glucose, LDL, and triglycerides (TG), at 35 days of age [63]. With an alternating dose of AFs in broilers chicks, poisoned with AFB_1_ (80 µg/kg/diet) during the first week of age and 520 µg/kg/diet in the remaining experimental period (other 4 weeks), resulting in a significant decrease in serum total protein, and an increase in ALP and ALT enzymes. ALT enzyme was significantly higher compared to the control and chicks fed with SIL or TB (3 g/kg/feed) supplementation, while AST was significantly lower in the group treated with AFB_1_ plus MT dried seed [45]. 

Some studies used lycopene, beta-glucans (extracted from *Saccharomyces cerevisiae* yeast), and sodium bentonite to compare the effects with MT. According to the study conducted by El-Sheshtawy and co-workers [61], on Pekin ducklings, ALT, AST, γ-GT, and ALP enzyme activities were significantly elevated in the Afs-intoxicated group (30 ppb/kg/diet, naturally contaminated). In comparison, they decreased significantly in groups treated with lycopene (100 mg/kg/diet) plus SIL (SIL 80%, analytical composition not reported) at a dose of 600 mg/kg-BW, as well as creatinine, with an associated increase of serum total protein and albumin. These latter parameters were altered in quail treated with AFB_1_ at a dose of 1500 μg/kg/diet [62], but not when fed with 250–500 mg/kg/diet of SIL [64]. No changes were observed in biochemical parameters in quail treated for 60 days with beta-glucans (1 kg/ton/diet) or SIL (500 g/ton/diet) plus AFB_1_ [62]. The administered dose of AFs (containing: AFB_1_: 84.64%; AFB_2_: 4.28%; AFG_1_: 11.07%) with any changes for AST, GGT, and CK values, showed that at this percentage, the additives were not able to mitigate the negative effects of the toxin on quail metabolism [62]. Otherwise, the combination of MT seed (0.5%/kg/diet) plus 0.5% sodium bentonite in the AFs-contaminated diet significantly increased the albumins level. At the same time, MT and sodium bentonite, separately or in combination, significantly decreased the AST level in AFs contaminated diet. Only the administration of MT seed plus AFB_1_ decreases the level of ALT and LDH [57]. In the trial conducted by Tedesco and co-workers [40] in AFB_1_-treated chickens, a lower level of ALT enzyme was observed. In SIL-treated chicks, the ALT enzyme did not decrease following AFB_1_ treatment. The serum levels of glucose, uric acid, AST, and ALT increased in response to OTA treatment (noted that glucose decrease usually in AFB_1_ intoxication), consistent with the common OTA intoxication [69]. The SIL treatment was found to protect against these increases [47]. Pigeons [65] treated with AFB_1_ by gavage for 2 consecutive days showed an increase in bile acid levels, AST, ALT, LDH, and creatine phosphokinase (CPK), with any variation of the levels of γ-GT. However, in the treatment with a high dose of SIL, these parameters were mostly unchanged. Furthermore, only pigeons treated with SIL at a dose of 10 mg/kg-BW, showed significant reductions in LDH, ALT, and CPK [65]. 

### 3.3. Influence of SIL on Serum and Tissues Antioxidant Parameters in MyT-Poisoned Poultry Species 

Of the studies analyzed, some of these investigate the antioxidant effects of SIL in MyT-poisoned poultry to evaluate its antioxidant potential, and the activity to restore the antioxidant defense of the tissues, in particular in the hepatic and muscle tissue. The study conducted by Egresi and co-workers [60], in ducks, evidenced that acute exposure to MyT (DON + ZEA) caused oxidative stress, which induced an effective antioxidant defensive response, indicated by the decrease in the hepatic MDA and diene conjugate content, not observed in the untreated group, and MT seed (0.50%/kg/diet) + MyT group. They observed also an increase in Al, Ca, Cu, and Fe, and a decrease in Mg, Mn, P, S, and Zn in both treated groups (MyT and MyT + MT). Contrary, always in ducks, was reported that on hepatic tissue there was a significant reduction in MDA level in the SIL (600 mg/kg-BW) plus MyT-treated group [61], showed also an improvement in the antioxidant levels of catalase (CAT), and glutathione S-transferase (GST) significantly decreased in the AFs-intoxicated group with a dose of 30 ppb/kg/diet (feed naturally contaminated).

As reported in an in vitro study, hepatocytes treated with OTA (1 µg/mL) for 24 h, reduced the superoxide dismutase (SOD), and GSH activity with an increase in MDA compared with the untreated cells [51]. The SIL treatment exhibited the largest antioxidation activity compared with the other three hepatoprotective agents tested in their study, glycyrrhizin, L-arginine, and glucorolactone [51]. At the same dose of OTA, in vivo study on intoxicated quails treated with SIL (500 mg/kg/diet), showed the highest total antioxidant capacity (TAOC), and GSH-Px and recorded the significantly lowest MDA values [64]. 

The antioxidant parameters were also measured in chicken muscle. The AFB_1_ treatment with a dose of 2 mg/kg/diet in *E. coli*-challenge broiler for 21 days, gave rise to the enhancement in thigh muscle’s MDA concentration, restored by the addition of MT dried seed (10 g/kg/diet) [54]. Following the effects on MDA concentrations, no differences in the thigh muscle’s ferric-reducing antioxidant power levels were noted in all experimental groups. Conversely, the study of Armanini and colleagues [48], reported no differences regarding the thiobarbituric acid reactive substances (TBARS) levels among the SIL-treated group with 100 mg/kg/diet (SIL 84.16% purity), plus AFB (0.05 ppb/kg/diet) + FUM (20 ppb/kg/diet), but showed a lower level of ROS in meat derived from the SIL-treated animals. The antioxidant defense systems of the animals are upregulated to counteract the excessive production of free radicals, including ROS observed in poisoned animals. Despite this, the addition of SIL restored ROS levels and GST activity evidencing the potent antioxidant capacity of its compound.

### 3.4. Influence of SIL on Immunological Parameters in MyT-Poisoned Poultry Species 

Immunosuppression in chickens can be caused by several factors such as natural, nutritional, managemental, diseases, stress, and vaccination, including also the interferences derived from MyT feed contaminations. MyT has a significant effect on poultry health, due to their interference with the immune response and response to the vaccination [70]. For example, OTA immunosuppression may be manifested as depressed T and B lymphocyte activity, suppressed immunoglobulin, antibody production [71], and complement activity in broiler chickens [72], including in the first life phase in ovo [73]. The same effects were noted in ducks when exposed to AFB_1_ [74]. On broiler, Fanni Makki and co-workers [49] investigated the trend of the Newcastle disease virus (NDV) or Avian Influenza (AI) antibody titers after 34 days of treatment with 250–500 ppb/kg/diet of AFB_1_ plus MT seed (0.5–1%/kg/diet). The administration of MT (at both doses) reduces the 500 ppb/kg/diet AFs effects with less reduction in the antibody titers, against NDV and AI. Chand and colleagues [44] found similar results studying the titers trends of NDV, infectious Bronchitis virus (IBV), and infectious Bursitis virus (IBv), evidencing that the contaminated diet with AFB_1_, significantly decreased serum mean antibody titers for these diseases. Again, the addition of MT seed at a dose of 10 g/kg/diet plus AFB_1_ protected from the reduction in humoral immune response in AFs poisoned broiler. Instead, in another study, the addition of MT seed powder at a dose of 10 g/kg/diet plus AFB_1_ increased the antibody titers (IgT) followed by the stimulation with the injection of sheep red blood cells (SRBC) in the broiler [55]. The administration of SIL at a dose of 1%/kg/diet showed also an increase in lysozyme concentration in animals poisoned by OTA, at concentrations very similar to the control group [58]. However, in Leghorn cockerels treated with SIL (10 g/kg/diet) in OTA contaminated diet (1 mg/kg/diet), no differences were noted for Ig titers at day 7 post-primary SRBC injection (p.i.) and 14 days p.i. The same effect was noted 24, 48, and 72 h p.i. after tubercolin administration [59]. The results obtained by Kathoon and co-workers [59] suggest that SIL possessed the ability to prevent titer decline caused by OTA-induced immunotoxic effects in chicks (1 mg/kg). Similarly, Raei and colleagues [54], showed the ability of the addition of 10 g/kg/diet of MT seed to counteract the decline of the total antibody titers in *E. coli*-challenged broiler. 

### 3.5. Influence of SIL on Performance and Products Quality Parameters in MyT-Poisoned Poultry Species

Mycotoxicosis inhibits poultry health, and consequently their performance, such as growth reducing feed consumption, digestion, absorption, enzymatic activity, and protein metabolism. In some studies, the MT potential to improve or prevent further worsening of the performance and quality parameters of products, such as poultry meat [29,75], was investigated when the performance can be worsened by MyT intoxication.

MT seed powder, whole plant powder, or an MT extract addition in feed, in MyT-intoxicated broiler, maintained a normal increase in BW compared to poisoned animals [56]. In this regard, the addition of MT seed powder (0.5%/kg/diet, analytical composition not reported) in an AFB_1_ (500 ppb/kg/diet)-contaminated diet, positively reduced the FCR compared to broiler fed with AFB_1_ alone, from 1 to 7 days of age. The addition of MT extract (600–1000 mg/kg/diet) had no significant effect on FCR. Overall, the addition of 1% of MT whole plant powder plus AFB_1_ in the diet proves to deliver the best results in terms of BW and FCR [56]. Chand and co-workers [44] reported an optimal BWG observed in the untreated broiler group, but also in the group poisoned with AFB_1_ (80–520 μg/kg/diet) plus MT seed powder (10 g/kg/diet, analytical composition not reported) showing the ability of MT to contrast the negative effects of AFs. These results on performance were similar to the findings by Tedesco and colleagues [40], with the administration of a dose of 600 mg/kg-BW of SIL-phytosome to an AFB_1_-poisoned broiler. They observed an increase in BW, compared to AFs-intoxicated broiler, and showed that SIL action was constant and positive to birds receiving only AFB_1_, considering BW and FI. These results suggest that treatment with SIL can be effective in counteracting the negative effects of AFB_1_ intoxication on FI and BW in growing broilers. In other studies, the BW was maintained with the administration of MT seed at a dose of 0.5–1%/kg/diet in a contaminated diet with AFB_1_ [53], and with MT seed powder at a dose of 10 g/kg/diet (silichristin 2851 mg/kg, silybin B 8864 mg/kg, SIL 100 mg/kg) [55]. Muhammad and co-workers [45] showed similar results, as an improved FI, BW, and best FCR using 10 g/kg/diet of dried MT seed added to AFB_1_ contaminated broiler diet. However, Raei and co-workers [54], evidenced no significant discrepancies between FI, BWG, and FCR. Furthermore, FCR was lower in the period between 1 to 7 days of age, for the MT-treated group (10 g/kg/diet), compared to the AFB_1_-poisoned group (2 mg/kg/diet). Even broilers exposed to a MyT mix (AFB + FUM) showed that SIL prevented the impairment of WG and FI compared to the MyT-treated group up to 21 days of the age of 42 days of trial, which evidenced that the addition of SIL minimizes the toxic effect of MyT on the broiler performance [48]. A dose ranging from 0.5 to 2 ppm/kg/diet of AFB_1_, resulted in a decrease in daily FI and DWG, with poor FCR, but the addition of 500 or 1000 ppm/kg/diet, especially at 1000 ppm (analytical composition not reported) of SIL contrast these effects, improving the WG, FI, and FCR [46]. In quails, a dose of 1000–2000 mg/kg/diet of SIL (analytical composition unknown) plus AFs (2.2 mg/kg/diet, as AFB_1_: 68.19%, AFB_2_: 4.57%, AFG_1_: 24.96%, AFG_2_: 2.28%), showing an increase in BWG correlated with a good FCR, with a minor rate of mortality compared to the poisoned group, and control group [63]. Similarly, Youssef and colleagues [64] evidenced an improvement, increasing supplementation levels of SIL to 500 mg/kg/diet plus AFB_1_ (19 ppb/kg/diet), in BW and BWG but not in FCR. Even in quails, the addition of 500 g/ton/diet of SIL in a diet contaminated with AFs (1500 μg/kg/diet, as AFB_1_: 84.64%; AFB_2_: 4.28%; AFG_1_: 11.07%), for 60 days of treatment, demonstrated that SIL was not able to mitigate the negative effects of AFs. In addition, the SIL plus AFB_1_-treated quail group produced 0.58% more eggs than the control group [62]. MT seed, at a dose of 5–10 g/kg added to the broiler diet containing AFB_1_ (250–500 ppb/kg), improved the ileal digestibility of dry matter, crude proteins, and Ca, evidenced that MT seed reduced the toxic effects of AFB_1_, facilitated the absorption of nutrients, and reduced the metabolic demands of the intestinal tract in broiler chickens [50]. Moreover, Armanini and co-workers [48] underline that the consumption of SIL affected the meat’s pH, increasing it. At the same time, the addition of SIL in a contaminated diet prevents cooking loss, showing a high water retention capacity and shear strength. Further, the PUFA was higher in the groups that consumed SIL and in the MyT+ SIL group. According to the study conducted by Jahanian and colleagues [46], the authors evidenced microbiota modifications; the ileal *E. coli*, *Salmonella* spp., *Klebsiella* spp. enumerations and total negative intestinal bacteria were markedly increased in poisoned chickens, both at 28 and 42 days of age, but not in the SIL-treated animals with a dose of 500 ppm/kg/diet. A strong suppressive effect of pathogenic microflora was evidenced in SIL plus AFB_1_ (0.5 ppm) group, with a dose of 1000 ppm of SIL, which may have contributed to indirectly improving the performance. 

### 3.6. Influence of SIL on MyT-Poisoned Livestock 

From the literature search, few studies on other farms or companion animals were detected, reporting similar positive effects observed in poultry species. In farm animals, a study was conducted on calves, and one on dairy cows. In calves [66] fed with 1.0 mg/kg AFB_1_ for 10 days, and subsequently treated with 600 mg/kg-BW of SIL (analytical composition not reported), and choline chloride (500 mg/kg-BW) for only 7 days, showed that the aflatoxicosis caused a worsening of the FI, and average daily weight gain (ADWG), associated with an increase in hepatic enzymes activity, such as AST, ALT, and renal function parameters, BUN and creatinine. The supplementation with SIL was comparatively more efficient to ameliorate the effects induced by AFB_1_ than choline chloride. With the use of an SIL-phytosome complex, SIL was effective in reducing the excretion of AFM_1_, the main metabolite of AFB_1_ toxin, in dairy cows. Tedesco and co-workers [67] investigated at the same time the use of SIL-phytosome and SIL standardized extract, at a dose of 30 g/day for 17 days, and 10 g/day (76% pure extract) for 9 days, respectively, in an organic dairy farm with high excretion of AFM_1_ in milk. In this farm, the results showed that the administered feed was naturally contaminated by AFB_1_. Both treatments do not completely inhibit AFM_1_ excretion in milk but contributed to reducing drastically its level. In other species such as the rabbit, raised as a pet, commercial choline (Epocler) plus SIL, turned out to be a valid association in the clinical treatment of AFB_1_ intoxication [68]. In a young female rabbit with ascites and hepatic damage caused by intoxication with 300 mg/kg/ feed of AFB_1_, the treatment for 1 month with SIL at a dose of 50 mg/kg/orally plus Epocler commercial product (1 mL/q) restore its hematological and biochemical parameters.

## 4. Conclusions

Cereals are the major ingredients for the nutrition of farm animals and provide primarily energy and proteins to the diet. One of the main concerns when using grain ingredients in feed formulation is MyT contamination. Overall, from the analyzed literature, several aspects emerged regarding the use of MT and the bioactive compound, SIL, against MyT toxicity, mainly evaluated in poultry species. The results, sometimes contradictory, display, in general, a broad range of MT/SIL effects finalized on the maintenance (due to its known important pharmacological properties) of animal health, restoring of the liver enzymes activity altered by the MyT-intoxication, reducing at the same time, the organs lesions caused by MyT. These latter effects, in many studies, were also supported and confirmed by histological evaluations. On immunity parameters, the treatment with MT can be effective in counteracting the negative effects of AFs intoxication with a decrease in antibody titers. On intestinal health, the use of MT in intoxicated animals showed an improvement in gut functionality. However, in some cases, the SIL supplementation does not produce appreciable effects to contrast ones caused by MyT. This can be due to the dose of MyT administered (range of administration), the time of exposition, and the dose of MT/SIL tested in the reported trials. Furthermore, it is necessary to underline the importance of the effective bioactive compounds content tested during MyT-poisoning. It is not clear if MT/SIL acts directly against MyT-intoxication, or if the positive effects observed can be related to the specific pharmacological properties of MT itself, for example acting as a hepatoprotector or as an antioxidant, and further trial can be useful. Nevertheless, the results showed that the use of MT/SIL, as a feed ingredient or feed additive, is a helpful natural dietary supplement to maintain the animals’ health, since the bioactive compounds, also if present in variable amounts, can help the animals to counteract the effects of MyT. The use of the phytoextract of SIL, due to its cost, can be useful if it is reported the specific bioactive compounds recognized for their pharmacological activities.

## Figures and Tables

**Table 1 animals-13-00330-t001:** Effects of milk thistle (MT) in chickens (*Gallus gallus*) poisoned by different types of mycotoxins (MyT). If not diversely reported in the section effects the results are referred to the control group without treatment.

Animal	MT (Type and Dose)	My (Type, Dose, and Route)	Exposure (Time)	Effects	Reference
Broiler (14-day-old)	SIL-Indena ^1^ (phytosome complex of SIL+phospholipids in a molar ratio of 1:2): 600 mg/kg-BW via gavage	AFB_1_: 0.8 mg/kg/diet	35 days	Performance AFB_1_: <*BW and FI. No difference in FCR in all groups. SIL group >**BW and FI. Biochemical parameters AFB_1_ group: <ALT. AFB_1_ + SIL group: no difference in ALT activity. Histopatology/organ lesions No change in liver weights in all groups. In AFB_1_-treated animals, the liver showed multifocal portal infiltration (mononucleates, granulocytes, and eosinophils) diffused in the parenchyma (portal areas), and necrosis. AFB_1_ + SIL group: <severe lesions.	[40]
in vitro model of the stomach and intestinal tracts of chickens (Erlenmeyer flask)	Esterified MT seed ^3^: 125–250 mg	AFB_1_ (from *A. flavus* colture): 250–500 μg/kg	37 °C for 3 h	Percentage of absorption ratio of AFB_1_ by MT seed at pH 4.5 to 6.5 Compared to respective positive control: At 250 µg/kg AFB_1_ only as a control: 0 At 500 µg/kg AFB_1_ only as a control: 0 At 125 mg of MT + 250 µg/kg AFB_1_: 30.14 ± 3.24 At 125 mg of MT+500 µg/kg AFB_1_: 26.15 ± 2.48 At 250 mg of MT + 250 µg/kg AFB_1_: 48.91 ± 3.69 At 250 mg of MT + 500 µg/kg AFB_1_: 41.39 ± 4.36	[43]
Broiler (1-day-old)	MT seed powder (MT): 10 g/kg/diet vs. TB (Mycoad): 3 g/kg/diet	Direct inoculation of *A. flavus* culture AFB_1_ producer in feed: 80–520 μg/kg/diet	5 weeks	Performance In control and MyT + MT group: >BW. BWG did not differ between the group treated with MT or TB. MT group: >BWG, water intake; <FCR. MyT + MT group: <FCR. Immunological parameters MyT group: <ND, IB, IBD antibody titers. MyT + MT group: >ND, IB, IBD antibody titers compared also to MyT + TB group. Histopatology/organ lesions MyT group: <thymus and bursa weight but no differences in other groups. MyT + MT group: weight of bursa restored and high weight of spleen compared to other groups.	[44]
Broiler (1-day-old)	MT dried seed: 10 g/kg/diet vs. TB (Mycoad): 3 g/kg/diet	AFB_1_ (from *A. flavus* colture): 80 µg/kg/diet (for 1 week) and 520 µg/kg/diet (for 4 weeks)	5 weeks	Performance AFB_1_ group: <BWG, FI, and FCR. MT + AFB_1_: >BWG, and FI. In all treatment groups: >FCR. MT + AFB_1_ group: better FCR compared to the others and control group. Biochemical parameters AFB_1_ group: <total protein; >ALP, ALT, AST. MT and TB group: <ALP, ALT, AST; >total protein. Histopatology/organ lesions AFB_1_ group: pale, enlarged (swollen), yellow friable livers with pinpoint hemorrhages, swollen kidneys, and atrophy of the bursa and thymus. MT + AFB_1_ group: <lesions induced by AFB_1_. TB group: no appreciable modifications in the lesions.	[45]
Broiler (7-day-old, male Ross 308)	SIL: 50–1000 ppm/kg/diet	AFB_1_ (from *A. parasiticus*): 0.5–2 ppm/kg/diet	42 days	Performance AFB_1_ group: <DFI, DWG, and >FCR. SIL + AFB_1_ group: >DFI, DWG, and <FCR. Histopatology/organ lesions AFB_1_ group: >heart weight. AFB_1_ (2 ppm) group: <in villi height, VH:CD, villi surface area, apparent villi absorptive area, and muscular diameter in jejunal sections. AFB_1_ + SIL group: any influence on organs weight. SIL (500 ppm) + AFB_1_ group: <crypt depth and goblet cell count; >villi height and width, VH:CD, villi surface area, apparent villi absorptive area, and muscular diameter. SIL (1000 ppm) + AFB_1_ (0.5 ppm) group: mitigated the depressed villi height, and VH:CD. Meat/carcass characteristics AFB_1_ group: <carcass yield AFB_1_ + SIL group: >carcass yield. Microbiota AFB_1_ group: >*E. coli*, *Salmonella*, *Klebsiella* count. SIL (500 ppm) group: <*E. coli*, *Salmonella*, *Klebsiella* count, and total negative bacteria. SIL (1000 ppm) + AFB_1_ (0.5 ppm) group: <count of negative bacteria.	[46]
Broiler (11-day-old, male Ross 308)	SIL (purity 60%): 1%/kg/diet	OTA (from *A. ochraceus* culture): 3 mg/kg/diet	32 days (broiler age 42 days)	Biochemical parameters OTA group: >glucose, uric acid, AST, and ALT. SIL group: <glucose, uric acid, AST, and ALT. SIL + OTA group: <AST, ALT. Organ lesions Only in the OTA group: small hemorrhages on the epicardium and duodenal mucosa, catarrhal enteritis. Kidneys and liver: congestion and enlargement. OTA + SIL group: no macroscopic lesions observed. Histopathology Compared to the OTA group in the OTA + SIL group: slight congestion of peritubular capillaries focal granular degeneration in the epithelial cells of convoluted tubules in kidneys, and depletion cells in the intestinal mucosa. In the liver, less cloudy swelling and granular or vacuolar degeneration of hepatocytes.	[47]
Broiler (1-day-old, Cobb 500)	Commercial SIL (SIL 84.16%): 100 mg/kg/diet	AFB (from *A. flavus* culture): 0.05 ppm/kg/diet + FUM (from *F. verticillioides* culture): 20 ppm/kg/diet	41 days	Performance MyT group: <WG, <FI, and >FCR. These effects were reduced by SIL. Biochemical parameters MyT group: >ALT, AST, uric acid. MyT + SIL group: <ALT activity. Histopatology/organ lesions No histological lesions in the liver and intestines of chickens in any group, but in the SIL group: >villus height and crypt depth. Meat/carcass characteristics In SIL + MyT group: <water cooking loss; >PUFA content (the same in SIL group).	[48]
Broiler (1-day-old, Ross 308)	MT seed: 0.5–1%/kg/diet	AFB_1_: 250-500 ppb/kg/diet	35 days	Performance AFB_1_ (250–500 ppb) group: <BW; >aggressive behavior, and disarray wings. Immunity After SRBC injection: no changes in antibody titer (NDV and AI) in any MT treatment group, but decreased in only AFB_1_ group. Histopatology/organ lesions AFB_1_ (500 ppb) group: <intestine length of the ileum and duodenum plus jejunum. AFB_1_ (250–500 ppb) + MT (0.5–1%): >intestine length of duodenum plus jejunum.	[49]
Broiler (1-day-old, Ross 308)	MT seed: 5–10 g/kg/diet	AFB_1_ (from *A. flavus* colture): 250–500 ppb/kg/diet	35 days	Digestibility of nutrients AFB_1_ (500 ppb) group: <ileal digestibility of dry matter, Ca, crude protein, and apparent digestible energy. MT (5–10 g) group: >digestibility of crude protein and Ca, AFB_1_ (250–500 ppb) + MT (5–10 g) group: no differences in Ca and crude protein digestibility. Histopatology/organ lesions AFB_1_ (500 ppb) group: <villus length, villus width, and VL/CD. AFB_1_ (250–500 ppb) + MT (5–10 g) group: no differences.	[50]
in vitro primary chicken hepatocytes	SIL 0.1–1–10 μg/mL	OTA:1 μg/mL	24 h At 37 °C	Cells response to OTA OTA: hepatocellular injury and >ALT, AST, MDA, mRNA expression of apoptosis-associated genes, and apoptosis rate; <SOD, and GSH levels. SIL treatment response SIL (0.1–1 μg/mL) + OTA: >cell variability, SOD, GSH (at 10 μg/mL SIL: <cell variability), ALT (no changes for AST activity); <MDA, mRNA expression of apoptosis-associated genes, and apoptosis rate.	[51]
Broiler (1-day-old, male Ross 308)	MT seed: 0–0.5–1%/kg/diet	AFB_1_ (from *A. flavus* colture): 250–500 ppb/kg/diet	21 days	Biochemical parameters AFB_1_ (250-500 ppb) group: <glucose, Ca, HDL, creatinine; >AST, ALT. MT (0.5–1%)+AFB_1_ (500 ppb) group: >glucose; <AST, and creatinine. MT (1%) + AFB_1_ (250–500 ppb) group: >HDL.	[52]
Broiler (1-day-old, male Ross 308)	MT seed: 0.5–1%/kg/diet	AFB_1_: 250–500 ppb/kg/diet	5 weeks	Biochemical parameters AFB_1_ (500 ppb) group: <albumin, direct bilirubin, Ca, and P; >uric acid, glucose, total bilirubin, ALT, AST, and γ-GT. MT (0.5–1%) + AFB_1_ group: <uric acid, glucose, AST, and γ-GT.	[53]
Broiler (1-day-old, male, Ross 308)	MT seed: 10 g/kg/diet	AFB_1_ (from *A. flavus* colture): 2 mg/kg/diet + *E. coli* challenge	21 days	Performance (before and after the challenge) AFB_1_ group: no difference in BWG and FI compared to other groups. MT + AFB_1_ group: <FCR, only from 0 to 7 days. Biochemical parameters (before and after the challenge) AFB_1_ group: <Ca and HDL; >ALT, AST. AFB_1_ + MT group: >Ca; <ALT, AST. Redox parameters (before and after the challenge) AFB_1_ group: >MDA in muscle. AFB_1_ + MT: <MDA in muscle. Immune system responses (before and after the challenge) No differences after the challenge.	[54]
Broiler (1-day-old, male Ross 308)	MTPowder ^2^: 10 g/kg/diet vs. TB (Toxofix-Arka): 1 g/kg/diet vs. *Spirulina platensis* (SP) 10 g/kg/diet	AFB_1_ (from an *A. parasiticus* culture): 0.6 mg/kg/diet	42 days	Performance AFB_1_ group_:_ <BW, and FI; >FCR. AFB_1_ + MT, AFB_1_ + TB, and AFB_1_ + SP group: >BWG, FI, but not in FCR. Biochemical parameters AFB_1_ group: >AST, and ALT. AFB_1_ + MT, AFB_1_ + TB, and AFB_1_ + SP group: <AST and ALT. Immunity AFB_1_ + MT group: <response to antibody titers against SRBC, and IgT titers compared to other groups. Microbiota AFB_1_ + MT, AFB_1_ + TB, and AFB_1_ + SP group: low *Coliforms* count.	[55]
Broiler (1-day-old, Ross 308)	MT seed powder: 0.5%/kg/diet plant powder: 1%/kg/diet extract: 600–1000 mg/kg/diet	AFB_1_: 500 ppb/kg/diet	35 days	Performance AFB_1_ group: <BWG; >FCR, and ALT. MT (1%) powder + AFB_1_ group: >BW; <FCR. MT seed powder (0.5%), plant powder (1%), and MT (1000 mg/kg) plant extract + AFB_1_ group: <ALT.	[56]
Broiler (1-day-old, male Ross 308)	MT seed: 0.5% vs. TB (Sodium Bentonite): 0.5%	AFB_1_: 500 ppm/kg/diet	4 weeks	Biochemical parameters AFB_1_ group: <albumin. AFB_1_ + TB group: >albumin. AFB_1_ + TB + MT: >albumin; <AST. MT + AFB_1_: <AST, ALT and LDH.	[57]
Broiler (1-day-old, male Ross 308)	SIL (purity 60%): 1% kg/diet	OTA (from *A. ochraceus* culture): 3 mg/kg/diet	42 days	Biochemical parameters OTA group: <lysozyme and beta-lysine concentration. SIL + OTA group: lysozyme concentrations, and beta-lysine activity not restored.	[58]
White Leghorn cockerel (1-day-old)	SIL 10 g/kg/diet + Vit. E: 200 mg/kg/diet	OTA (from *A. Ochraceus* culture): 1–2 mg/kg/diet	42 days	Immunity Antibody titers against SRBC injection OTA (2 mg) + SIL: <titers. IgG titers in OTA (2 mg) + Vit. E, OTA (2 mg) + SIL, and OTA (2 mg) + Vit. E + SIL group: significant differences. At 14 days post-p.i. OTA alone (1 mg), OTA (2 mg) + Vit. E, OTA (2 mg) + SIL, and OTA (2 mg) + Vit. E + SIL group: <total Ig titers. OTA (1–2 mg), OTA (2 mg) + Vit. E, OTA (2 mg) + SIL, and OTA (2 mg) + SIL + Vit. E group: <titers.	[59]

^1^ silybin 49.1%, isosilibin 14.3%, silydianin 14.6%, silychristin 8.3%, taxifolin 4.3%; ^2^ silychristin 2851 mg/kg, silybin B 8864 mg/kg, silymarin 100 mg/kg; ^3^ taxifolin 2.42 ± 0.02 mg/g-DW(dry weight), silychristin 2.28 ± 0.02 mg/g-DW, silydianin 4.31 ± 0.04 mg/g-DW, silybin A 1.25 ± 0.009 mg/g-DW, silybin B 3.55 ± 0.008 mg/g-DW, isosilibin A 2.45 ± 0.01 mg/g-DW, isosilibin B 2.72 ± 0.02 mg/g-DW. *<: poor, worse, decrease; **>; good, improve, increase.

**Table 2 animals-13-00330-t002:** Effects of milk thistle (MT) in other poultry species, poisoned by different types of mycotoxins (MyT). If not diversely reported in the section effects the results are referred to the control group without treatment.

Poultry Species	MT (Type and Dose)	MyT (Type, Dose, and Route)	Exposure (Time)	Effects	Reference
Duck	MT seed (Safimpex, commercial product): 0.50%/kg/diet	DON: 4.9 mg/kg/diet + ZEA: 0.66 mg/kg/diet	47 days	Histopatology/organ lesions	[60]
				MyT group: **>grade vacuolar degeneration of hepatocytes cytoplasm, necrosis, and cell deaths of the mononuclear phagocyte system. Focal lymphocytic and histiocytic interstitial infiltrates and mild interstitial fibrosis.	
				MT + MyT group: <*vacuolar degeneration of hepatocytes.	
				Redox parameters	
				MyT group: <MDA, diene conjugate, and free sulfhydryl.	
				MT + MyT group: >DC, MDA, and free sulfhydryl.	
				Trace elements	
				In MT + MyT and MyT group: >Al, Ca, Cu, Fe; <Mn, P, Zn, S content.	
*Anas platyrhynchos domesticus* (Pekin duckling, 1-day-old)	SIL 80%: 600 mg/kg-BW vs. Lycopene (LyC) 20 mg (LYC-O-MATO commercial product): 100 mg/kg/diet	AFs: 30 ppb/kg/diet (naturally contaminated)	24 days (2 weeks AFs exposure + 10 days of aflatoxicosis treatment)	Biochemical parameters	[61]
				AFs group: >ALT, AST, γ-GT, ALP, creatinine; <total protein, and albumin.	
				SIL + AFs group: <ALT, AST, γ-GT, ALP, and creatinine; >total protein and albumin.	
				Redox parameters	
				AFs group: >MDA; <TOAC, GST, and catalase activity.	
				SIL + AFs group: <MDA; >TOAC, GST, and catalase activity.	
*Coturnix coturnix japonica* (Japanase quail, 12 week-old)	SIL 84.91%: 500 g/ton/diet vs. Beta-glucans, extracted from *Saccharomyces cerevisiae* yeast: 1 kg/ton/diet	AFB_1_ (from *A. parasiticus* culture) ^1^: 1500 μg/kg/diet	60 days	Performance	[62]
				AFB_1_ group: <FI.	
				AFB_1_ + SIL group: >0.58% eggs produced.	
				Biochemical parameters	
				AFB_1_ group: >AST, GGT, CK levels.	
				SIL + AFB_1_ group: <GGT.	
*Coturnix coturnix japonica* (7-day-old, broiler Japanese quail)	SIL: 1000–2000 mg/kg/diet	AFs ^2^: 2.2 mg/kg/diet	35 days	Performance	[63]
				AFs group: <FI, DWG, >mortality.	
				SIL (1000–2000 mg/kg) + AFs group: >BWG; <FCR.	
				Biochemical parameters	
				AFs group: >ALT, ALP, uric acid; <total protein, creatinine, and Ca.	
				SIL (2000 mg/kg) + AFs group: <ALT, AST, ALP, and glucose; >P.	
*Coturnix coturnix japonica* (1-day-old)	SIL: 250–500 mg/kg/diet	AFB_1_: 19 ppb/kg/diet (naturally contaminated)	35 days	Performance	[64]
				AFB_1_ + SIL (250–500 mg/kg) group: >BWG, BW but no in FI and FCR.	
				Biochemical parameters	
				AFB_1_ + SIL (250–500 mg/kg) group: no differences in total plasma proteins, albumins, and globulins; <AST.	
				Antioxidant parameters	
				AFB_1_ + SIL (500 mg/kg) group: >TAOC and GSH-Px; <MDA in liver tissue.	
				Carcass characteristics	
				AFB_1_ + SIL group: <AFs residues in tissues.	
*Columba livia* White Carneaux pigeon (12 months of age)	SIL 80%: 10–100 mg/kg-BW	AFB_1_: 3 mg/kg-BW by gavage for 2 consecutive days	Diet for 21 days and then continue until the end of the experiment (day 60)	Histopatology/organ lesions	[65]
				AFB_1_ group: >hepatic inflammation and necrosis, biliary-duct hyperplasia, and lymphocyte infiltration.	
				SIL group: the liver injury was not significantly affected by SIL treatment.	
				Biochemical parameters	
				SIL (10 mg/kg) + AFB_1_ group: <ALT, CPK, LDH, creatinine.	

^1^ AFs content: AFB_1_: 84.64%; AFB_2_: 4.28%; AFG_1_: 11.07%; ^2^ AFs content: AFB_1_: 68.19%, AFB_2_: 4.57%, AFG_1_: 24.96%, AFG_2_: 2.28%. *<: poor, worse, decrease; **>; good, improve, increase.

**Table 3 animals-13-00330-t003:** Effects of milk thistle (MT) in other animal species, poisoned by different types of mycotoxins (MyT). If not diversely reported in the section effects the results are referred to the control group without treatment.

Categories and Specie	MT (Type and Dose)	MyT (Type, Dose, and Route)	Exposure (Time)	Effects	Reference
Ruminant: Calve (6–12 month of age)	SIL: 600 mg/kg-BW + choline chloride 500 mg/kg orally for 7 days	AFB_1_: 1.0 mg/kg-BW for 10 days, daily, through gelatinized capsules.	10 days	Performance	[66]
				AFB_1_: <*FI, ADWG	
				AFB_1_ + SIL: >**FI, ADWG better than AFB_1_ + choline.	
				Biochemical parameters	
				AFB_1_: <blood cell count; >AST, ALP, BUN, and creatinine.	
				AFB_1_ + SIL: >blood cell count; <AST, ALP, BUN, and creatinine better than choline treated group.	
Ruminant: dairy cow (Italian Friesian)	SIL (76% Indena standardized extract) ^1^: 10 g/day/cow + SIL + phytosome (molar ratio 1:2): 30 g/day/cow via oral drench	AFB_1_ (feed naturally contaminated): 0.80 ± 0.2 µg/kg (1st treatment) AFB_1_ (feed naturally contaminated): 0.44 ± 0.3 µg/kg (2nd treatment).	1st treatment: SIL for 9 days 2nd treatment: SIL+ phytosome for 17 days	AFM_1_ milk excretion (1st treatment)	[67]
				For the whole period: <AFM_1_ (in particular on day 3 of the treatment).	
				AFM_1_ milk excretion (2nd treatment)	
				AFM_1_: <(constant) from day 0 to 17 (in particular at day 11) in treated animals.	
*Oryctolagus cuniculi* (pet rabbit, clinical case of AFB intoxication)	SIL: 50 mg/kg/orally + Epocler: 1 mL/q (choline)	AFB_1_: 300 mg/kg/feed (naturally contaminated)	12–24 h	Clinical evidence of AFB_1_ intoxication	[68]
				ascites (with sterile exudate); >AST and ALT.	
				WBC in normal value, but macrocytic RBC was evidenced.	
				Clinical evidence after SIL treatment	
				restores hepatic activity at normal parameters.	

^1^: silybin 49.1%, isosilibin 14.3%, silydianin 14.6%, silychristin 8.3%, taxifolin 4.3%. *<: poor, worse, decrease; **>; good, improve, increase.

## Data Availability

Not applicable.

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
