# Peer review of "Restoring Activity of Milk Thistle (Silybum marianum L.) on Serum Biochemical Parameters, Oxidative Status, Immunity, and Performance in Poultry and Other Animal Species, Poisoned by Mycotoxins: A Review"

_animals, 2023, doi:10.3390/ani13030330_

Round 1
Reviewer 1 Report
I have some suggestions to improvement that have been written below.
Line 129: this is review paper no need to write methodology!
Line 162: Authors should start discussion with topics that they mentioned in line 175 instead of writing result and discussion.
Line 250: It is better to mention Table in one place!
Line 267 - 271: this sentence is not clear ‘’ In a study conducted by ……..'' it is better to rewrite again.
- Authors have to increase some new studies about this topic because there are many old papers appear in this manuscript. Authors can cite some new papers which are relate with that topic for example the paper which is written below:
Bendowski, W., Michalczuk, M., Jóźwik, A., Kareem, K. Y., Łozicki, A., Karwacki, J., & Bień, D. (2022). Using Milk Thistle (Silybum marianum) Extract to Improve the Welfare, Growth Performance and Meat Quality of Broiler Chicken. Animals, 12(9), 1085.
Regards,
Author Response
Reviewer 1
I have some suggestions to improvement that have been written below.
Line 129: this is review paper no need to write methodology!
Reply. Thank you for your suggestion. We deleted this line.
Line 162: Authors should start the discussion with topics that they mentioned in line 175 instead of writing results and discussion.
Reply. Thank you for your suggestion. As requested, we changed the numbering and started the discussion section with the topic titles.
Line 250: It is better to mention Table in one place!
Reply. As requested and to simplify, the mention of tables in the text was removed. However, it was indicated only once in the results section, in lines 169-171.
Line 267 - 271: this sentence is not clear ‘’ In a study conducted by ……..'' it is better to rewrite again.
Reply. Thank you for your suggestion. The requested change at lines 267-271 was done, and the paragraph was rewritten. It should be clearer.
Authors have to increase some new studies about this topic because there are many old papers appear in this manuscript. Authors can cite some new papers which are relate with that topic for example the paper which is written below:
Bendowski, W., Michalczuk, M., Jóźwik, A., Kareem, K. Y., Łozicki, A., Karwacki, J., & Bień, D. (2022). Using Milk Thistle (Silybum marianum) Extract to Improve the Welfare, Growth Performance and Meat Quality of Broiler Chicken. Animals, 12 (9), 1085.
Reply. Thanks for the suggestion. The study indicated was cited at lines 417-419.

Reviewer 2 Report
General comments:
The manuscript “Restoring Activity of Milk Thistle (Silybum marianum L.) on Serum Biochemical Parameters, Oxidative Status, Immunity, and Performance in Poultry and Other Animal Species, Poisoned by Mycotoxins: A Review” is of an important and actual topic. The research summarizes the knowledge of the use of milk thistle on health and performances of poultry and other livestock.
The text is fluently written, the approach used is clearly presented, and includes only minor typos, and it is easy to read. I have only minor comments for the authors to clarify before I can recommend this manuscript for publication.
Minor comments:
L10: please delete toxic.
L11: replace ‘animal farm’ with ‘livestock’
L33: replace ‘farming’ with’nutrition’
LL37-38: In general, I’d go for words that are not included in the title. For example I suggest to use: Silymarin; Phytoextract, Feed additives, Bioactive compound, Animal health.
L89 and elsewhere: please use italics for Latin words.
Author Response
Reviewer 2
The manuscript “Restoring Activity of Milk Thistle (Silybum marianum L.) on Serum Biochemical Parameters, Oxidative Status, Immunity, and Performance in Poultry and Other Animal Species, Poisoned by Mycotoxins: A Review” is of an important and actual topic. The research summarizes the knowledge of the use of milk thistle on health and performances of poultry and other livestock.
The text is fluently written, the approach used is clearly presented, and includes only minor typos, and it is easy to read. I have only minor comments for the authors to clarify before I can recommend this manuscript for publication.
Reply.
Thank you for your appreciation of the manuscript and for your suggestions.
Minor comments:
L10: please delete toxic.
Reply. Thank you for your suggestion. The requested change was done.
L11: replace ‘animal farm’ with ‘livestock’
Reply. Thank you for your suggestion. The requested change was done.
L33: replace ‘farming’ with ’nutrition’
Reply. Thank you for your suggestion. The requested change was done.
LL37-38: In general, I’d go for words that are not included in the title. For example I suggest to use: Silymarin; Phytoextract, Feed additives, Bioactive compound, Animal health.
Reply. Thank you for your suggestion. The requested change was done.
L89 and elsewhere: please use italics for Latin words.
Reply. The use of "Asteraceae" not in italics is due to nomenclature. Indicating only the plant family, the use of italics is formally accepted.

Reviewer 3 Report
This article intended to provide a review on the efficacy of milk thistle to counteract mycotoxins toxicity on organs, biochemical and immunological functions, and performance in poisoned poultry and livestock. It is interesting. The major problem is the total number for the related research articles is only 27, which seemed rather limited for a review. Subsequently, the protect effects observed from these limited studies are questionable. Additionally, the mechanisms should be also reviewed, such as the gene clusters related to the protect effects, to make the article more helpful for the researchers.
Author Response
Reviewer 3
This article intended to provide a review on the efficacy of milk thistle to counteract mycotoxins toxicity on organs, biochemical and immunological functions, and performance in poisoned poultry and livestock. It is interesting. The major problem is the total number for the related research articles is only 27, which seemed rather limited for a review. Subsequently, the protect effects observed from these limited studies are questionable. Additionally, the mechanisms should be also reviewed, such as the gene clusters related to the protect effects, to make the article more helpful for the researchers.
Reply.
Dear reviewer, thank you for your consideration and opinion about our manuscript. We agree with your opinion, however, in scientific non-experimental studies (review and others) on chicken or other animal species, the effect that milk thistle and its derivatives have against mycotoxins is often generically reported probably referring to, as reported by European Medicines Agency (EMA) that the component of milk thistle extract, stimulated the protein biosynthesis and induced a membrane stabilization of hepatic cells that may prevent toxins transport. Despite this, from our investigation, specific experimental works on the evaluation of the effects of MT and derivatives in animals resulted in only 27. However, we believe it is important to report the studies, precisely because the antitoxic effect of silymarin is currently widely recognized by the scientific community.
Regarding the mechanism of action, we add in text: Detailed information on the multifunctional and multitarget activity and the potential mechanism of action of MT and derivative products were reported in the EMA report, and by Tedesco and Guerrini (Use of Milk Thistle in Farm and Companion Animals: A Review, 2022). Regarding genes, an old work by Sonnenbichler & Zeti (1992), reported that Silibinin, a component of silymarin, did not induce new genes but could stimulate DNA replication and mitosis if the signal for cell proliferation and mitosis was given. In this regard, EMA reports that MT, among its various properties, can influence the gene expression of some genes (antioxidants, anti-inflammatories, etc.) as also reported in our recent review (Tedesco and Guerrini, 2022) Undoubtedly there is some evidence of MT interaction with genes and their expression, although none specifically concerning mycotoxins.

Round 2
Reviewer 3 Report
Ok for me.